# Postnatal Cytomegalovirus Infection May Increase the Susceptibility of Tuberous Sclerosis Complex to Autism Spectrum Disorders

Xiao-Yan Yang,[a,b] Yang-Yang Wang,[b] Yue-Peng Zhou,[c] Jing He,[d] Meng-Jie Mei,[c] Meng-Na Zhang,[a,b] Bin Wang,[e] Wen-Jing Zhou,[d] Min-Hua Luo,[c] Qiu-Hong Wang,[a,b] Zhong-Yuan Li,[f] Yong Xu,[a,b] Qian Lu,[a,b] Li-Ping Zou[a,b,g]

aMedical School of Chinese PLA, Beijing, China

bFaculty of Pediatrics, Chinese PLA General Hospital, Department of Pediatrics, The First Medical Center of Chinese PLA General Hospital, Beijing, China

cWuhan Institute of Virology, Chinese Academy of Sciences, Wuhan, China

dDepartment of Neurosurgery, Yuquan Hospital of Tsinghua University, Beijing, China

eDepartment of the Outpatients, The First Medical Center of Chinese PLA General Hospital, Beijing, China

fFaculty of Pediatrics, Chinese PLA General Hospital, Department of Pediatrics, The Fourth Medical Center of Chinese PLA General Hospital, Beijing, China

gBeijing Institute for Brain Disorders, Center for Brain Disorders Research, Capital Medical University, Beijing, China

**ABSTRACT** Autism spectrum disorder (ASD), a highly hereditary and heterogeneous neurodevelopmental disorder, is influenced by genetic and environmental factors. Tuberous sclerosis complex (TSC) is a common syndrome associated with ASD. Cytomegalovirus (CMV) infection is an environmental risk factor for ASD. The similarities in pathological and mechanistic pathways of TSC and CMV intrigued us to investigate whether CMV and TSC interacted in ASD's occurrence. We detected CMV IgG seroprevalence of 308 TSC patients from our prospective cohort (September 2011 to March 2021) and 93 healthy children by magnetic particle indirect chemiluminescence immunoassay. A total of 206 TSC patients enrolled were divided into ASD and non-ASD groups, and the relationship between ASD and CMV seroprevalence was analyzed. Nested PCR and Western blot were used to detect CMV DNAs and proteins in cortical malformations of seven TSC patients with and without ASD. No difference was found in CMV seroprevalence between TSC patients and healthy children (74.0% versus 72.0%, $P = 0.704$). Univariate analysis showed the seroprevalence in TSC patients with ASD was higher than that in TSC patients without ASD (89.2% versus 75.1%, $P = 0.063$), and multifactorial analysis showed that CMV seroprevalence was a risk factor for ASD in TSC patients (OR = 3.976, 95% CI = 1.093 to 14.454). Moreover, CMV was more likely to be detected in the cortical malformations in TSC patients with ASD but not in those without ASD. The findings demonstrated that CMV may increase the susceptibility of TSC to ASD.

**IMPORTANCE** CMV is an environmental risk factor for ASD, but its role in syndromic autism with known genetic etiology has been rarely studied. The pathogenesis of ASD is related to the interaction between environmental and genetic factors. This study demonstrated that CMV can contribute to the occurrence of ASD related to TSC, a common genetic syndrome associated with ASD. Our findings provided support for the theory of gene-environment interaction (G × E) in pathogenesis of ASD and a new perspective for the prevention and therapy for TSC related ASD.

**KEYWORDS** cytomegalovirus, autism spectrum disorder, gene-environment interaction, tuberous sclerosis complex

Address correspondence to Li-Ping Zou, zouliping21@hotmail.com.

The authors declare no conflict of interest.

Autism spectrum disorder (ASD) is a common childhood neurodevelopmental disorder characterized by persistent impairments in communication and social interactions as well as restricted and repetitive patterns of behavior, interests or activities. The

global prevalence of ASD is about 1/132, which represents the main psychiatric cause of disability in children under 5 years old (1). However, the pathogenesis of ASD is still unclear. ASD is a highly hereditary and heterogeneous disease, which may be affected by gene-environment interactions (G × E). Many possible candidate genes and chromosomal abnormalities have been discovered, but the discrepancy is substantial (2); and environmental risk factors may play a reactive, independent, or contributory role in the occurrence of ASD (3–5). In recent years, a large-scale multinational cohort study found that environmental factors had different effects on ASD among people in different countries, suggesting that different genetic backgrounds might have different susceptibility to the environmental factors (6).

However, current studies have been focused on exploring the single etiology of ASD while ignoring the additive effects of other factors. In particular, few studies have attempted to investigate the contribution of environmental factors on ASD syndrome caused by a certain gene. Fortunately, our prospective tuberous sclerosis complex (TSC) cohort and clinical specimen library provided an opportunity to investigate the gene-environment interactions in syndromic ASD. TSC, an autosomal-dominant hereditary disease caused by either TSC1 or TSC2 gene mutations, is one of the major syndromes accompanied by ASD. The prevalence of ASD in TSC ranges from 17% to 63% (7), which is influenced by some factors, such as early onset epilepsy, refractory epilepsy, infantile spasm, and neurodevelopmental retardation (7–10). Regarding the environmental risk factors, cytomegalovirus (CMV) infection was related to ASD, as revealed by researchers more than 40 years ago (11), this finding was supported by a number of subsequent studies (12–16). It is noted that most of these studies have focused on congenital CMV infections, while few studies have investigated the role of postanal CMV infection on ASD due to most postanal infections being asymptomatic. Nevertheless, there are studies showed that postnatal CMV infection had adverse effects on both general intelligence during early childhood in general populations and overall cognitive functions in adolescents born preterm (17, 18), suggesting that postnatal CMV infection also plays a role in neurodevelopment, especially on the basis of neurodevelopmental defects. Therefore, although postnatal CMV infection is often asymptomatic, its effect on ASD is worth studying.

Interestingly, congenital CMV infection and TSC have similar intracerebral pathologies in addition to their association with ASD. Both can cause similar cortical malformations, periventricular calcification, and abnormally enlarged neurons (2, 19–21). The pathogenesis mechanism may be due to the fact that both are related to the overactivation of mTOR pathway. The pathogenesis of TSC is that heterozygous mutations in the either TSC1 gene or TSC2 gene lead to overaction of mTOR signaling pathway, resulting in excessive cell growth and the formation of hamartomas in multiple systems throughout the body (2). CMV relies on the mTOR pathway, especially PI3K/AKT/mTOR pathway, to entry host cells and optimize conditions for replication and allowing for the infected cells to persist long enough for the virus to complete lytic replication (22–25). In addition, PI3K activation is needed to establish the latency, but the specific role of mTOR during latency is not clear yet (26, 27). Moreover, in recent years, studies have shown that mTOR pathway is involved in the pathogenesis of ASD, whether in syndromic ASD or in idiopathic ASD without specific genetic etiology (28).

The similarity of CMV and TSC inspired us to investigate the relationship between CMV and ASD in TSC patients. We aimed to explore whether postnatal CMV infection can enhance the pathogenic effect of TSC on ASD, hoping that our study can provide clinical support for G × E theory of the pathogenesis of ASD and provide a new horizon for ASD management.

## RESULTS

**TSC patients are not more susceptible to CMV.** No significant difference in CMV seroprevalence was found between 308 patients with TSC and 93 healthy children (74.0% of TSC patients versus 72.0% of healthy children, $P = 0.704$) (Fig. 1A). No

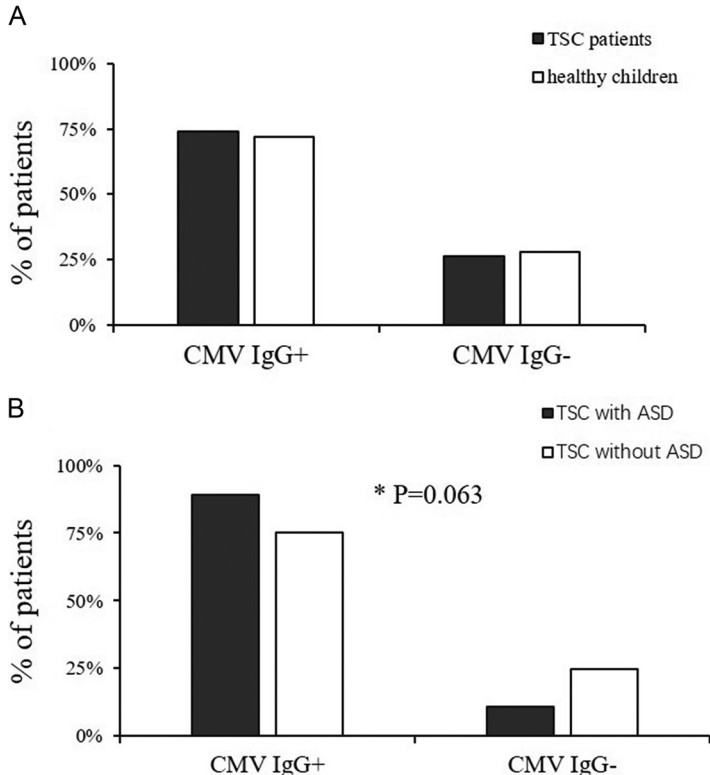

**FIG 1** The CMV seroprevalence of each group. (A) There was no difference in CMV seroprevalence between TSC patients and healthy children. (B) The CMV seroprevalence of TSC patients with ASD was higher than that of patients without ASD (P = 0.063). *Considering the unidirectional impact of CMV on ASD, one-side test and a P-value of 10% were used.

significant difference was found with regard to age (3.50 (4.98) years old of TSC patients versus 3.91 (0.42) years old of healthy children, P = 0.266) and gender distribution (P = 0.249) between two groups.

**CMV seroprevalence is a risk factor for patients with TSC and ASD.** Diagnostic and Statistical Manual of Mental Disorders (DSM-5) assessments for TSC children were available for 206 patients in our cohort, including 37 diagnosed with ASD and 169 without ASD (Fig. 2). Among 37 ASD patients, 89.2% occurred in CMV seropositive patients, and 10.8% occurred in the seronegative group. Patients with TSC and ASD (TSC/ASD) had a higher seroprevalence (89.2%) than those without ASD (75.1%), and the difference was statistically significant (P = 0.063, one-side test) (Fig. 1B). In addition, significant differences were found in neurodevelopmental retardation (yes or no, P < 0.001) and refractory seizures (yes or no, P = 0.010) between two groups. Patients with TSC/ASD did not differ significantly from those without ASD with regard to female gender, infantile spasms (yes or no), and seizure onset age (Table 1).

Forced entry binary logistic regression was performed with seven independent variables, which were chosen from their statistical significance on the univariate analysis of ASD in TSC and possible risk factors reported in other studies (7–10). These seven variables were CMV IgG serostatus (positive or negative), gender, neurodevelopmental retardation (yes or no), infantile spasms (yes or no), refractory epilepsy (yes or no), age of seizure onset, and age of ASD assessment. Gene mutation was not included in this analysis because a considerable number of patients (31.6%) had not completed this testing. The results showed that CMV IgG seroprevalence was a risk factor for ASD in TSC patients (OR = 3.976, 95%CI = 1.093 to 14.454) (Fig. 3).

**CMV DNAs can be more likely detected in the cortical malformations in TSC patients with ASD.** According to the clinical records, three patients (4, 6, and 7) were diagnosed with TSC/ASD and four patients without ASD (1, 2, 3, 5) (Fig. 4). The nested

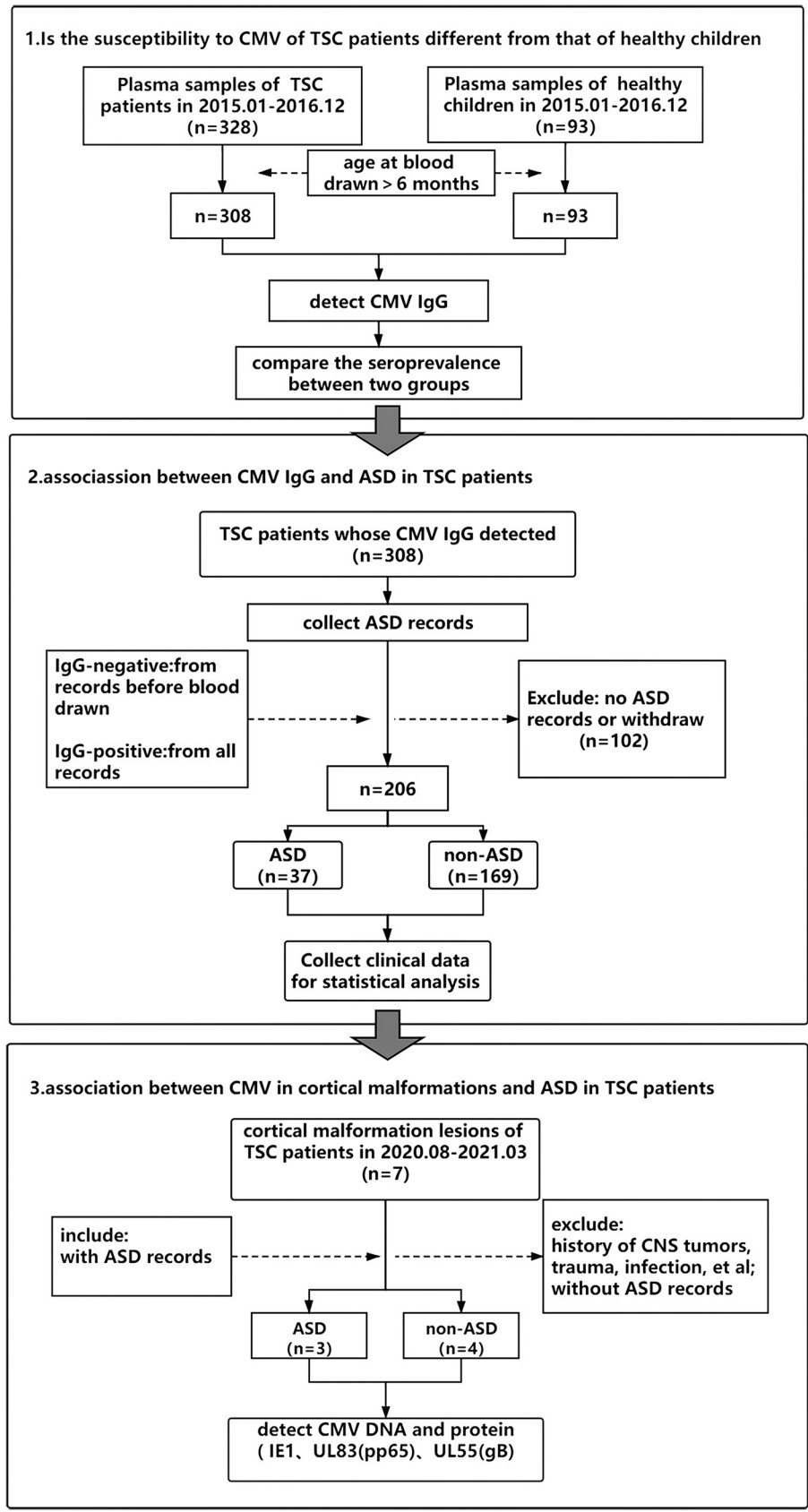

**FIG 2** The workflow of this study.

**Table 1** Main clinical characteristics of TSC patients with and without ASD

| Items | TSC/ASD | Non-ASD | P |
|---|---|---|---|
| No. | 37 | 169 | |
| Sex(female) | 13 (35.1%) | 76 (45.0%) | 0.274 |
| Age of seizure onset (months) | 8.00 (7.00) | 10.00 (19.25) | 0.122 |
| Age of ASD evaluation (years) | 4.75 (4.58) | 4.92 (5.29) | 0.936 |
| CMV IgG positive | 33 (89.2%) | 127 (75.1%) | 0.063[b] |
| Mental retardation | 36 (97.3%) | 60 (35.5%) | <0.001 |
| | | | |
| Epilepsy | | | |
| With | 35 (94.6%) | 154 (91.1%) | 0.743 |
| Infantile spasms | 14 (37.8%) | 48 (28.4%) | 0.257 |
| Refractory epilepsy | 23 (65.7%) | 64 (41.6%) | 0.010 |
| | | | |
| TSC2 gene mutation[a] | 23 (88.5%) | 81 (70.4%) | 0.059 |

[a]A total of 141 patients completed genetic testing.
[b]CMV was a risk factor for ASD; therefore, one-side test and a p-value of 10% were used. P<0.05 was considered statistically significant on other analysis. TSC, tuberous sclerosis complex; ASD, autism spectrum disorder; TSC/ASD, TSC patients with ASD; Non-ASD, TSC patients without ASD; CMV, cytomegalovirus.

PCR (nPCR) results showed that the IE1, UL55, and UL83 DNAs were all strong positive in patients with TSC/ASD. Among patients without ASD, patient 1 and patient 2 seemed weak positive for UL55 DNA and IE1 DNA, respectively, patient 5 was strong positive for IE1 DNA, and patient 3 had negative results of three DNAs. As for the Western blot (WB) results, no protein was clearly detected in samples despite of several extremely faint signals of IE1 protein. These results showed that CMV DNAs were more likely to be detected in the cortical malformations of TSC patients with ASD, but our sample size was too small to draw a conclusion.

## DISCUSSION

Given the high heritability and heterogeneity of ASD, the theory of gene-environment interaction in ASD has been widely recognized. However, no individual-level evidence can support this theory, especially for the monogenic cause of ASD, where little attention has been paid to the contribution of environmental factors (5). Our research has proved for the first time that even for autism caused by a single gene, specific environmental factors play an additive role in autism occurrence.

We found that seropositive of CMV was a risk factor for ASD in TSC patients by uni-

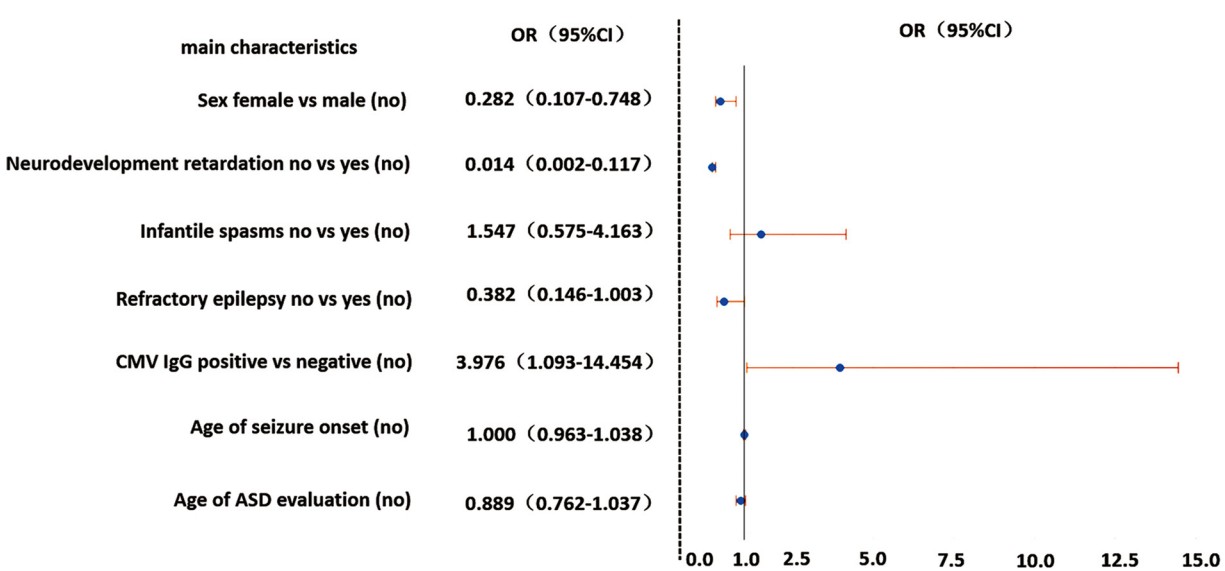

**FIG 3** The OR and 95% CI of each variable in the logistic model. OR, odds ratio; CI, confidence interval.

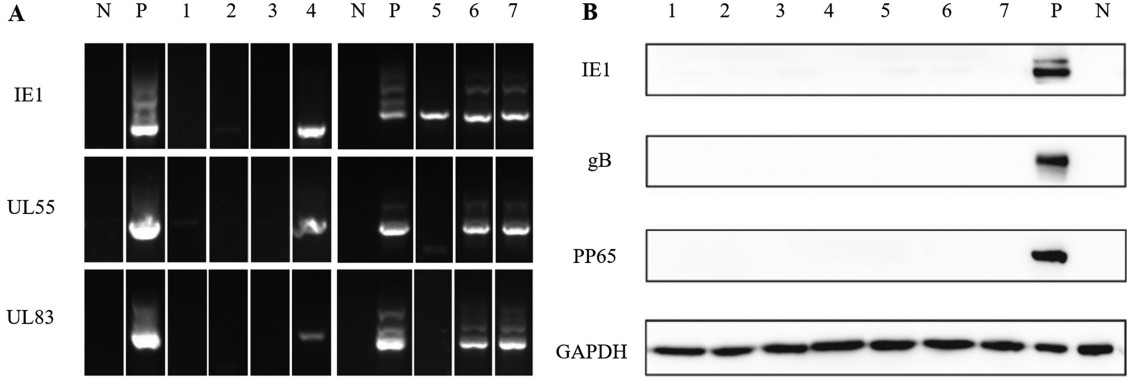

**FIG 4** The nPCR and WB results of CMV in cortical malformation lesions of TSC patients. IE1, UL55, and UL83 DNAs encode proteins IE1, gB, and pp65 respectively. Numbers 4, 6, and 7 represent TSC patients with ASD; numbers 1, 2, 3, and 5 represent TSC patients without ASD. (A) nPCR results. In TSC patients with ASD, three DNAs were all positive; in TSC patients without ASD, only patient 5 had a clear positive result of IE1, patient 1 had a weak positive for UL55, and patient 2 had a weak positive for IE1. (B) WB results. No CMV proteins were clearly detected in the lesions of all patients. nPCR, nested PCR; WB, Western blot; N, negative control; P, positive control; GAPDH, reduced glyceraldehyde-phosphate dehydrogenase, a kind of housekeeping protein.

variate and multivariate analysis. Regarding the results of the detection of CMV DNAs and proteins in the brain, it seemed that CMV DNAs were easier to be detected in the brain of TSC/ASD patients, which provided a support for the seroprevalence result. However, among patients without ASD, one had a strong positive for IE1, and two had a weak positive for one kind of DNA, respectively. The possible reason that only one type of DNA was detected in a patient (either strong positive or weak positive) was that the abundances of the other two types DNAs were too low to be detected, which suggested that CMV might have little impact on the brain because the content was low. Different from previous studies that did not find the relationship between CMV seroprevalence and ASD (29, 30), our studies at plasma-level demonstrated that CMV was a risk factor for ASD related to TSC. The discrepancy between studies can actually reinforce our finding because the previous studies neither looked specifically at ASD in TSC patients, while we specialized in TSC patients, which means we controlled the confounding factor of different genetic etiology compared with previous studies. Therefore, the inconsistency between studies can prove that the genetic background of TSC and CMV have interaction in ASD; in other words, CMV makes the genetic background of TSC more susceptible to ASD.

As for the specific mechanism of CMV promoting ASD in TSC patients, there might be some possibilities. Indirective effect of the inflammation or immune response caused by CMV is thought to play a role, as shown by a study that the TSC2 haploinsufficiency and gestational immune activation synergize to disrupt adult social approach behavior in mice (31). Although this study focused on the immunity during pregnancy, it at least suggested that immunity could be a possible mechanism for ASD. Notably, TSC/mTOR signaling also plays a role in immune modulation apart from the role in neurodevelopment, ASD, and CMV infection (32). As an important signaling hub, mTOR pathway can response to multiple cellular and environmental cues (33) and it has been studied that there are some mechanism crossover points between CMV and TSC, for example, the UL38 protein of CMV can antagonize the TSC protein complex (34) and CMV can change the substrate specificities and rapamycin sensitivity of mTORC1, which is the target of mTOR inhibitor rapamycin in the treatment of TSC (35). Hence, the impact of CMV on mTOR pathway as well as the clinical consequences of TSC are worth studying.

Previous studies on the relationship between ASD and CMV mainly focused on the congenital infection field because this period is critical for neurodevelopment and susceptible to be damaged by CMV infection. In contrast, our study applied CMV IgG that only represented the infection history in which most cases would be postnatal infected. Although fewer postnatal infections are symptomatic, the impact of postnatal infection on ASD should not be neglected. It is worth noting that, in addition to the fetal period, the first year of life is also a critical period for brain development, when

more neurons continue to migrate to the outermost layer of the cortex, thereby significantly increasing cortical gray matter volume (106%) (36, 37). Coincidentally, most postnatal CMV infections occur within 1 year of age. We proposed that the primary infection or repeated reactivation of CMV may affect the brain structural development and functional connectivity of patients with TSC and ASD, although the intermittent and cyclic effect of CMV on human was difficult to monitor. The frequency and degree of virus reactivation may affect the severity of its effect on the neuropathological change and patients' clinical manifestation.

This study has limitations. First, CMV IgG can only reflect whether, not when, the patient has been infected, which is determined by the characteristics of CMV infection itself; that is, most postnatal infections are asymptomatic and the diagnosis is always conducted retrospectively through serum antibodies. But this does not affect our results, because we aimed at studying the relationship between postnatal CMV infection history and ASD. Second, serological sample size was not large enough, especially in the case group, which mainly resulted from the intrinsic characteristic of low prevalence of ASD in our TSC cohort. In addition, the number of brain specimens was also too small to obtain conclusion but rather only provide simple verification for the seroprevalence study and indication for CMV's effect on TSC/ASD may be due to its role in brain. We have been keeping collecting specimens and will conduct further research on this field. Third, the patients tested for serum IgG and those tested for intracerebral viruses were not from the same group, so the patients tested for intracerebral CMV did not purposefully test IgG in advance. However, this study did not evaluate the relationship between CMV seroprevalence and intracerebral infection but rather the effect of CMV on ASD. Finally, we did not include gene mutation in the regression analysis because a certain number of patients did not complete the gene test due to heavy economic burden or ignorance of the cause due to the mildness of symptoms. This resulted in an inability to assess the relationship between CMV and TSC1/TSC2 genes, respectively, but it did not affect the main results.

In conclusion, our study primarily showed that postnatal CMV infection contributed to the occurrence of ASD in TSC populations, which supported the theory of certain genetic ASD subpopulations having different sensibility to specific environmental risk factors. Stratified studies of ASD subgroups based on genetic etiology may help to reveal the gene-environmental interactions in ASD. We hope to see more investigators cover this field of research to elucidate the etiology mechanisms and to make a therapeutic breakthrough of ASD at the molecular level.

## MATERIALS AND METHODS

**Participants.** Since September 2011, we have established a prospective cohort for TSC (Registration No. CHICTR-OCH-13003763, ESOSIPT Trial) consisting of TSC child patients from all over China who were admitted to the Department of Pediatrics of The First Medical Center of Chinese PLA General Hospital. All patients in this cohort were dynamically registered with clinical manifestation data, diagnosis, and treatment records by experts. The typical manifestations of TSC patients include epilepsy, ASD, neurodevelopment retardation, skin lesions, and cardiac/kidney hamartomas, and so on (21). The clinical data of patients with TSC enrolled in the present study came from this cohort. During this cohort's construction process, we collected peripheral blood samples from 328 TSC patients and 93 healthy children who underwent physical examination in our hospital from January 2015 to December 2016. After centrifugation, the upper plasma was frozen in the −80°C refrigerator to establish a clinical blood specimen bank. Blood was drawn from the 308 TSC children and 93 healthy children of ages greater than 6 months old who were included in this study (Fig. 2). The reason for limiting the age is that the IgG antibodies in children aged 6 months old or less are a mixture of self-produced and maternal antibodies, which would interfere with the determination of whether the child has been infected with CMV.

In addition, some patients in the TSC cohort underwent surgery for refractory epilepsy. We collected surgical specimens from seven TSC patients who were treated jointly by our team and the Department of Neurosurgery of Yuquan Hospital of Tsinghua University from August 2020 to March 2021 to establish a cortical malformation lesion sample bank of TSC (Fig. 2). After the surgical removal of the lesions, a specimen of about 1 cm$^3$ was immediately separated and blood was rinsed with normal saline, then frozen in liquid nitrogen and transferred into a refrigerator (–80°C) for storage within 48 h.

**Study design.** This is a case-control study. TSC patients who underwent plasma CMV IgG testing were divided into ASD and non-ASD groups based on the diagnosis records according to Diagnostic and Statistical Manual of Mental Disorders (DSM-5) criteria. The diagnosis of ASD was made by the chief expert in our cohort study (38). Considered the facts that CMV IgG serostatus represented the infection history and every patient had multiple ASD records, we used different strategies to select ASD records

**Table 2** Primer sequences and amplification conditions of CMV genes

| Gene | Primer | | Sequence | Temp (°C) | Fragment (bp) |
|------|--------|---|----------|-----------|---------------|
| IE1 | OUTER | F | TCCTCTGCCAAGAGAAAGATGGACC | 57 | 1,647 |
| | | R | TCTCAGACACTGGCTCAGACTTGAC | | |
| | INNER | F | GACATGGTGCGGCATAGAATCAAGG | 55 | 400 |
| | | R | CATTGGTGGTCTTAGGGA | | |
| UL83 | OUTER | F | GACGCGTCAGCAGAACCAGTGGAAA | 55 | 1,058 |
| | | R | AGATGTCGTTGGCGTCCCAGAAGAA | | |
| | INNER | F | TTTACCTCACACGAGCATTTTGGGC | 60 | 527 |
| | | R | TCCTCGTCGGTGTCCTCTT | | |
| UL55 | OUTER | F | AACAAACCGATTGCCGCGCGTTTCA | 57 | 856 |
| | | R | AAACACTTTCCTCGTAGGAAGGCGG | | |
| | INNER | F | ACCACCGCACTGAGGAAT | 57 | 563 |
| | | R | GGACACCAGATAGGGAAAGA | | |

on the basis of CMV IgG results. When the IgG was negative, we judged whether the patient had ASD according to the records before blood drawing to ensure that the ASD records were collected under the context of not being infected by CMV. When the IgG was positive, in order to reduce the possibility of losing information, we judged from all records regardless of the time of blood drawing (the workflow is shown in Fig. 2). In addition, data on gender, age, gene mutation (TSC1 or TSC2), neurodevelopment retardation (yes or no), and history of epilepsy, including infantile spasms (yes or no), refractory epilepsy (yes or no), and age of seizure onset, were also collected.

In order to provide supporting evidence for seroprevalence study, we detected CMV infection in cortical malformations of TSC patients. The patients whose brain lesion samples had been collected were divided into ASD and non-ASD groups according to their clinical records, and nPCR and WB were used to detect the DNAs and proteins of CMV.

**Detection of plasma CMV IgG.** The concentrations of CMV IgG in the plasma of 308 TSC patients and 93 healthy children were detected by magnetic particle indirect chemiluminescence immunoassay. According to the instruction of the reagent (LIAISONR CMV IgG II), the CMV IgG was negative if the concentration was less than 12.0 U/mL, which indicated no CMV infection. If the concentration was higher than 14.0 U/mL, then the CMV IgG was positive, thereby indicating a past infection. If the result was between 12.0 and 14.0 U/mL, then it was considered suspicious, and a second test was performed.

**Detection of DNAs and proteins of CMV in the cortical malformation lesions of TSC patients.** After thawing the seven brain specimens, nested PCR was used to detect the immediate early gene IE1 and early genes UL83 and UL55 of CMV. Western blot was used to detect the corresponding proteins IE1, pp65, and gB (37). The specific methods were as follows.

Nested PCR: the primer sequences and amplification conditions of the three genes are shown in Table 2. The total genomic DNA of brain tissue was extracted using Tiangen Genome Extraction Kit (TIANGEN Biotech, China). In the first round of PCR, extracted tissue DNA (200 ng) was added as a template, and the outer primers were used for amplification. In the second round of PCR, the first round's products (2 $\mu$L) were used as the templates, and the corresponding inner primers were added for amplification. Finally, the PCR products of the second round were run on electrophoresis gel, and the presence or absence of the gene was confirmed according to the band size and sequencing.

Western blot: the brain tissue was detached in phosphate buffer saline (PBS) and centrifuged at 1,500 rpm for 5 min. Then, the cell pellets were lysed in radioimmunoprecipitation assay (RIPA) buffer and crushed ultrasonically (43% power, 8 times, 3 s each time) into non-sticky status. Then, they were centrifuged at 12,000 rpm at 4°C for 5 min. After determining the protein concentration by Biyuntian BCA kit, equal amounts of cell lysates were separated by sodium dodecyl sulfate-polyacrylamide gel electrophoresis (SDS-PAGE) and transferred to a polyvinylidene difluoride (PVDF) membrane (Millipore). They were incubated with the indicated primary and corresponding secondary antibodies, and then, signals were detected by using a chemiluminescence instrument and analyzed by densitometry (ImageJ; National Institutes of Health). At least three sets of independent experiments were performed. CMV proteins were detected using mouse monoclonal antibodies to IE1 (Virusys corporation P1215), gB (Virusys P1201), and pp65 (made by the laboratory of Wuhan Institute of Virology) and a rabbit polyclonal antibody to GAPDH (Proteintech 10494-1-AP). The secondary antibodies used were horseradish peroxidase (HRP) conjugated goat anti-mouse IgG (H+L).

**Ethical considerations.** Informed consent was obtained from the parents or guardians for the collection of blood and brain samples. This study was approved by the Ethics Committee of the Chinese PLA General Hospital and the Ethics Committee of Yuquan Hospital of Tsinghua University.

**Statistical analysis.** SAS 9.4 was used for statistical analysis. Chi-square test or Fisher's exact test was used for qualitative data. Independent-sample $t$ test or Mann-Whitney U test was used for quantitative data. We reported the data as mean ± standard deviation (SD) or as median and interquartile range (IQR), respectively. Forced entry binary logistic regression was used to assess the impact of some specific factors on the likelihood of having ASD in our TSC cohort. The exponentiation of the B coefficient values was used to determine odds ratios. A $P$-value of 5% in the two-sides test was considered statistically significant, but in the univariate analysis of the CMV seroprevalence in the ASD and non-ASD groups, one-

side test and a *P*-value of 10% were used, because according to clinical knowledge and literature reports, CMV was a risk factor for ASD and its impact on ASD was unidirectional.

**Data availability.** The data of this study are available on request from the corresponding author. The data are not publicly available due to privacy or ethical restrictions.

## ACKNOWLEDGMENTS

We thank all the children of our TSC cohort as well as healthy children who agreed to provide blood samples, and their parents or guardians. We are also grateful to the neurosurgeons of the third ward of the Department of Neurosurgery of Yuquan Hospital of Tsinghua University for their help in collecting the cortical malformation samples. Finally, we also thank the Department of Microbiology of Chinese PLA General Hospital for helping to detect plasma CMV IgG.

This research was supported by the National Key Research and Development Program of China (No. 2016YFC1000707) and the Capital Health Research and Development of Special (2022-1-5081).

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
