## [Reviewer comments · Microbiology Spectrum]

Microbiology Spectrum

Postnatal Cytomegalovirus Infection May Increase the Susceptibility of Tuberos Sclerosis Complex to Autism Spectrum Disorders

xiaoyan Yang, Yang-Yang Wang, Yue-peng Zhou, Jing He, Mengjie Mei, Mengna Zhang, Bin Wang, Wenjing Zhou, Min-Hua Luo, Qihong Wang, Zhongyuan Li, Yong Xu, Qian Lu, and Li-Ping Zou

Corresponding Author(s): Li-Ping Zou, Chinese PLA General Hospital

Review Timeline:

Submission Date:	October 12, 2021
Editorial Decision:	November 3, 2021
Revision Received:	December 13, 2021
Editorial Decision:	February 18, 2022
Revision Received:	March 6, 2022
Accepted:	April 1, 2022

Editor: Clinton Jones

Reviewer(s): The reviewers have opted to remain anonymous.

Transaction Report:

DOI: <https://doi.org/10.1128/spectrum.01864-21>

November 3, 2021

Dr. Li-Ping Zou
Chinese PLA General Hospital
Department of Pediatrics
28 Fuxing Road
Beijing
China

Re: Spectrum01864-21 (Cytomegalovirus Infection Contributes to the Occurrence of Autism Spectrum Disorders Associated with Tuberous Sclerosis Complex)

Dear Dr. Li-Ping Zou:

Thank you for submitting your manuscript to Microbiology Spectrum. When submitting the revised version of your paper, please provide (1) point-by-point responses to the issues raised by the reviewers as file type "Response to Reviewers," not in your cover letter, and (2) a PDF file that indicates the changes from the original submission (by highlighting or underlining the changes) as file type "Marked Up Manuscript - For Review Only".

It is crucial for you to address each Major Concerns raised by both reviewers. In particular, both reviewers raised concerns about whether the statistical analysis, sample size, and methodology, which are central to the manuscript's conclusions. If the Major Concerns are not adequately addressed, the manuscript will be rejected.

Please use this link to submit your revised manuscript - we strongly recommend that you submit your paper within the next 60 days or reach out to me. Detailed information on submitting your revised paper are below.

Link Not Available

Sincerely,

Clinton Jones

Journals Department
Reviewer comments:

Reviewer #1 (Comments for the Author):

This manuscript entitled "Cytomegalovirus Infection Contributes to the Occurrence of Autism Spectrum Disorders Associated with Tuberous Sclerosis Complex" by Yang et al. describes a cohort study of patients suffering from tuberous sclerosis complex (TSC) disorder with and without autism spectrum disorder (ASD). Based on neuropathological similarities of TSC and congenital CMV infection, the authors wanted to investigate whether CMV infection is a risk factor for ASD in TSC patients. They found that prior CMV infection (as measured by serum anti-CMV IgG) did not correlate with a significantly increased risk for ASD ($P=0.063$), but a multifactorial analysis showed that CMV seroprevalence might be a risk factor for ASD in TSC patients ($OR=3.976$, 95% CI 1.093-14.454). The authors also detected CMV DNA by nested PCR in cortical malformations of three TSC+ASD patients. The authors concluded that CMV contributes to the occurrence of ASD in TSC patients.

There are several issues regarding study design and interpretation of data that need to be addressed.

Questions and concern

- 1) The authors emphasize similarities of neuropathological symptoms in TSC and congenital CMV infection, both of which can be associated with ASD. However, they did not investigate the prevalence of congenital CMV infection, but simply determined the CMV serostatus. The vast majority of CMV infections are postnatal, and postnatal infections are only rarely associated with CNS infection and damage. This raises the question whether serum anti-CMV IgG is an appropriate parameter.
- 2) There is no significant correlation between CMV serostatus and ASD in TSC patients ($P=0.063$). The odds ratio appears to be slightly increased in a multifactorial analysis, but the authors did not describe in the Materials & Methods section how this analysis was done. Moreover, a correlation does not prove causality. Hence it remained unclear whether CMV is a risk factor for ASD or ASD is a risk factor for acquiring CMV.
- 3) The number of brain samples analyzed is way too small to draw any meaningful conclusions. The detection of CMV DNA by nested PCR can have many reasons, and the failure to detect CMV proteins by Western blot may also have many reasons, including low sensitivity of the WB. One cannot conclude from the data presented that the CMV is present in a latent form. It is also not mentioned which of the brain samples are from CMV-positive and which from CMV-negative patients.

Reviewer #2 (Comments for the Author):

This study by Yang et al. utilizes data and samples collected in China from children with TSC and healthy controls to investigate a link between CMV and ASD in TSC patients. The authors conclude that CMV seroprevalence, but not a number of other variables (sex, neurodevelopment retardation, infantile spasms...), was a risk factor for ASD in TSC patients.

Major Points:

- 1) The authors reference a number of studies linking HCMV to the TSC/mTOR signaling pathway. However, these papers all refer to a lytic infection. These studies conclude that HCMV modulates this pathway to optimize conditions for replication and to allow for the cell to persist long enough for the virus to complete lytic replication, but do not investigate latency or long-term persistence of any form. Thus, the comment on lines 81-82 "...thereby achieving long-term existence in the human body" is misleading and not supported by the provided references.
- 2) The presence of IE1 DNA, but not UL55 or UL83, in patient 5 seems unexpected. No explanation for this was provided. Also, given a clear positive result for IE1 in patient 5, is the statement on lines 119-120 "No CMV DNA and protein were found in the brain lesions of the four patients without ASD" correct?
- 3) No conclusions can be made from blank western blots without a positive control for each of the antibodies. Therefore, a positive control lane must be included and run on the same blot as the samples to make any conclusions about lack of viral protein.
- 4) On lines 137 to 142, the authors discuss an apparent discrepancy between their studies and two other published studies in which no association between CMV and ASD was found. This discussion is somewhat confusing, given that neither study looked specifically at ASD in TSC patients. The authors should reevaluate the discussion of these papers, which may actually reinforce their findings that CMV contributed to ASD in TSC patients, given that TSC/ASD is only a subset of all ASD cases.

Minor Points:

- 1) Figure references do not appear in order in the text (Fig2A precedes Fig1)

Staff Comments:

Preparing Revision Guidelines

Please return the manuscript within 60 days; if you cannot complete the modification within this time period, please contact me. If you do not wish to modify the manuscript and prefer to submit it to another journal, please notify me of your decision immediately so that the manuscript may be formally withdrawn from consideration by Microbiology Spectrum.

Dear reviewers ,

First, we would like to thank you for the positive and constructive comments and suggestions concerning our manuscript. We have substantially revised our manuscript point-by-point (by highlighting the changes in marked-up manuscript). Besides, we have made some adjustments to the structure of the Introduction, Method and Discussion sections, mainly changing the position of some sentences or paragraphs and simplifying some sentences (by underlining the changes in marked-up manuscript) without changing the original meaning.

Follows are our responses to your comments.

Reviewer #1:

(1) The authors emphasize similarities of neuropathological symptoms in TSC and congenital CMV infection, both of which can be associated with ASD. However, they did not investigate the prevalence of congenital CMV infection, but simply determined the CMV serostatus. The vast majority of CMV infections are postnatal, and postnatal infections are only rarely associated with CNS infection and damage. This raises the question whether serum anti-CMV IgG is an appropriate parameter.

Response: Thank you for your question. We acknowledge that the retrospective analysis of the relationship between anti-CMV IgG and ASD has limitation, which are discussed at the end of the discussion section : anti-CMV IgG can only reflect whether a person has been infected with CMV in the past but not the specific time of infection, moreover, most cases are asymptomatic postnatal infections (lines183-188).

Despite the limitation, there are rationalities of studying the relationship between ASD and CMV through serum IgG. We introduced this in the Introduction section on lines 72-79 and in Discussion section on lines 169-182. **First of all**, though postnatal CMV infection rarely causes obvious brain

abnormalities, the mechanism of ASD is complex and unclear, and may be related not only to the direct brain damages but also the indirect inflammation or immune response. **Secondly**, there are studies shown that postnatal CMV infection has adverse effects on both general intelligence during early childhood in general populations and overall cognitive functions in adolescents born preterm (1,2), suggesting that postnatal CMV infection play a role in the neurodevelopment, especially on the basis of neurodevelopmental defects. **Thirdly**, studies investigating the influence of CMV on ASD mainly focused on congenital infection, as this period is critical for the neurodevelopment. What is noteworthy is that the first year of life is still an important period for brain development (3,4), which coincidentally is also a period of susceptibility to CMV. But there was one limitation of our study was that we cannot ascertain whether postnatal CMV infection occurred within the first year of age. **Finally**, CMV will exist in the body lifelong after primary infection and will be reactivated opportunistically, so its impact on the human body is dynamic, long-term and difficult to monitor. To sum up, the influence of postnatal CMV infection on ASD is worth studying.

Although the similar neuropathological symptoms (similar brain damages and both associated with ASD) caused by congenital CMV infection and TSC inspired us to conduct this study, we aimed to investigate the relationship between postnatal CMV infection history and ASD rather than the association between ASD and brain damage caused by congenital CMV infection. We emphasized this on lines 93-94, 199-200.

1. Lee S M, Mitchell R, Knight J A, et al. Early-childhood cytomegalovirus infection and children's neurocognitive development[J]. *Int J Epidemiol*, 2021, 50 (2): 538-549. DOI:10.1093/ije/dyaa232.
2. Brecht K F, Goelz R, Bevot A, et al. Postnatal Human Cytomegalovirus Infection in Preterm Infants Has Long-Term Neuropsychological Sequelae[J]. *Journal of Pediatrics*, 2015, 166 (4): 834-U122. DOI:10.1016/j.jppe

ds.2014.11.002.

3. Marin O, Valiente M, Ge X, et al. Guiding neuronal cell migrations[J]. CSH Perspect Biol, 2010,2 (2): a001834. DOI:10.1101/cshperspect.a001834.

4. Gilmore J H, Shi F, Woolson S L, et al. Longitudinal development of cortical and subcortical gray matter from birth to 2 years[J]. Cereb Cortex, 2012, 22 (11): 2478-2485. DOI:10.1093/cercor/bhr327.

(2) There is no significant correlation between CMV serostatus and ASD in TSC patients (P=0.063). The odds ratio appears to be slightly increased in a multifactorial analysis, but the authors did not describe in the Materials & Methods section how this analysis was done. Moreover, a correlation does prove causality. Hence it remained unclear whether CMV is a risk factor for ASD or ASD is a risk factor for acquiring CMV.

Response : Thanks for your questions. **First**, the original statistical test in univariate analysis obtained $P=0.063$ based on the two-sides test, but considering CMV is an environmental risk factor for ASD, one-side test and a p-value of 10% are appropriate. Therefore, the difference between CMV serostatus and ASD in TSC patients was statistically significant. We modified the corresponding part in the revision (lines 34, 108-109, 141-142, 294-297, 429-430). **Second**, we also noted the odds ratio was slightly increased, but it did not mean our result was not powerful. The small OR value was due to the low prevalence of ASD in our TSC cohort (17.96%). Besides, TSC is a typical syndrome with ASD and the prevalence of ASD in TSC is reported to be about 17-63%, while the incidence of ASD in our CMV-seronegative group is only 8.7%, which is much lower than that of external controls. This also supports the result of our research. **Third**, as for the causality, our retrospective study cannot determine the timing sequence of CMV infection and ASD occurrence, so it is only allowed to determine the correlation between CMV and TSC/ASD

rather than the causality by statistical methods. However, clinical knowledge can help to confirm that CMV is a risk factor for ASD, which has been studied in various studies. We performed this study because there had been no research investigating the role of CMV in syndromic ASD caused by a specific genetic etiology, and our research was conducive to studying the gene-environment interaction in the occurrence of ASD.

As for the multifactorial analysis method, we modified this section on Materials & Methods and Result sections (lines 113, 291-293). Forced entry binary logistic regression was used. The dependent variable is whether or not ASD. The seven independent variables were CMV IgG serostatus (positive or negative), gender, neurodevelopmental retardation (yes or no), infantile spasms (yes or no), refractory epilepsy (yes or no), age of seizure onset, and age at ASD assessment, which chosen from their statistical significance on the univariate analysis in our study (neurodevelopmental retardation and refractory seizures) and possible risk factors reported in other studies. The exponentiation of the B coefficient values was used to determine odds ratios. A p-value of 5% in the two-sides test was considered statistically significant.

(3) The number of brain samples analyzed is way too small to draw any meaningful conclusions. The detection of CMV DNA by nested PCR can have many reasons, and the failure to detect CMV proteins by Western blot may also have many reasons, including low sensitivity of the WB. One cannot conclude from the data presented that the CMV is present in a latent form. It is also not mentioned which of the brain samples are from CMV-positive and which from CMV-negative patients.

Response: Thank you for your comment. The brain samples were too small to allow us to draw a certain and reliable result of the correlation between CMV infection in the brain and ASD, which was one of the limits of our study. However, TSC is a relative rare disease with an incidence of 1/6000, and only

a part of them is eligible for surgery, which make it difficult to collect enough brain samples from surgery in a short time. We have been keeping collecting specimens and will conduct further research on this field. Considering the low statistical efficiency of small sample size, we only aimed to provide supporting evidence for the results of CMV seroprevalence study by investigating the CMV infection in the brain lesions of TSC patients with or without ASD. We have made corresponding explanations on lines 130-133, 188-190, 244-245.

As for the problem of latent infection, thank you for your correction, it is incorrect and too decisive to judge whether CMV is in latent period only by PCR and WB results. We have removed the description of latent infection from corresponding sections and only objectively describe the positive and negative results of DNA and protein tests. (lines 37-38, 122-133, 258-260, 437-439)

The patients whose brain samples were collected were not from the same sample bank as those who were tested CMV serum IgG. The former came from the serum specimen bank established from January 2015 to December 2016, and the latter came from the brain specimen bank established by August 2020 to March 2021 (lines 214-217, 224-226). Because specimens were collected retrospectively from the banks, patients whose brain samples were collected were not purposefully tested for CMV serum antibodies previously. This was one of the limitations of our study, but our study did not evaluate the relationship between CMV seroprevalence and intracerebral infection but rather the effect of CMV on ASD. We described this part on Discussion section of limitations on lines 190-194.

Reviewer #2:

Major Points:

(1) The authors reference a number of studies linking HCMV to the TSC/mTor signaling pathway. However, these papers all refer to a lytic infection. These studies conclude that HCMV modulates this pathway to

optimize conditions for replication and to allow for the cell to persist long enough for the virus to complete lytic replication, but do not investigate latency or long-term persistence of any form. Thus, the comment on lines 81-82 "...thereby achieving long-term existence in the human body" is misleading and not supported by the provided references.

Response: Thank you for your correction. I am sorry that the role of mTOR pathway in CMV infection was not clearly and correctly described according to the references. We have revised this part as follows and the references according to your comments (lines 86-90): CMV relies on the mTOR pathway, especially PI3K/AKT/mTOR pathway, to entry host cells (4,5) and optimize conditions for replication and allowing for the infected cells to persist long enough for the virus to complete lytic replication, besides, PI3K activation is needed to establish the latency but the specific role of mTOR during latency is not clear yet (6,7).

4. Altman AM, Mahmud J, Nikolovska-Coleska Z, Chan G. 2019. HCMV modulation of cellular PI3K/AKT/mTOR signaling: New opportunities for therapeutic intervention? *Antiviral Res* 163:82-90.

5. Cobbs C, Khan S, Matlaf L, McAllister S, Zider A, Yount G, Rahlin K, Harkins L, Bezrookove V, Singer E, Soroceanu L. 2014. HCMV glycoprotein B is expressed in primary glioblastomas and enhances growth and invasiveness via PDGFR-alpha activation. *Oncotarget* 5:1091-1100.

6. Buehler J, Zeltzer S, Reitsma J, Petrucelli A, Umashankar M, Rak M, Zagallo P, Schroeder J, Terhune S, Goodrum F. 2016. Opposing Regulation of the EGF Receptor: A Molecular Switch Controlling Cytomegalovirus Latency and Replication. *PLoS Pathog* 12:e1005655.

7. Kim JH, Collins-McMillen D, Buehler JC, Goodrum FD, Yurochko AD. 2017. Human Cytomegalovirus Requires Epidermal Growth Factor Receptor Signaling To Enter and Initiate the Early Steps in the Establishment of Latency in CD34 + Human Progenitor Cells. *J Virol* 91:e01206-16.

(2) The presence of IE1 DNA, but not UL55 or UL83, in patient 5 seems unexpected. No explanation for this was provided. Also, given a clear positive result for IE1 in patient 5, is the statement on lines 119-120 "No CMV DNA and protein were found in the brain lesions of the four patients without ASD" correct?

Response: Thanks for your question. We thought the reason for the only presence of IE1 but not UL55 or UL83 in patient 5 might be the gene abundances of UL55 and UL83 in tissues were too low to be detected. In addition, when we reviewed our result, we found that among patients without ASD, patient 1 seemed weak positive for UL55 DNA and patient 2 seemed weak positive for IE1 DNA. Only one type of DNA was detected in a patient (either strong positive or weak positive) suggested that CMV might have little impact on the brain because the content was low. We described and discussed this part on lines 126-129, 144-148.

Sorry for the incorrect statement of "No CMV DNA and protein were found in the brain lesions of the four patients without ASD", we have modified the description of the result of CMV detection in the brain samples (lines 122-133). In addition, according to the comment of reviewer 1, we deleted the original description of latent infection, and only objectively described the results of PCR and WB, because it was decisive and incorrect to judge latent infection only based on the result of PCR and WB.

(3) No conclusions can be made from blank western blots without a positive control for each of the antibodies. Therefore, a positive control lane must be included and run on the same blot as the samples to make any conclusions about lack of viral protein.

Response: Thanks for your suggestion. We have performed WB again with positive control and changed the corresponding figure as follow.

(4) On lines 137 to 142, the authors discuss an apparent discrepancy between their studies and two other published studies in which no association between CMV and ASD was found. This discussion is somewhat confusing, given that neither study looked specifically at ASD in TSC patients. The authors should reevaluate the discussion of these papers, which may actually reinforce their findings that CMV contributed to ASD in TSC patients, given that TSC/ASD is only a subset of all ASD cases.

Response: Thanks for your suggestion, we rediscussed this section on lines 149-155: Different from previous studies that did not find the relationship between CMV seroprevalence and ASD (29, 30), our studies at plasma-level and brain-level jointly proved that CMV infection promoted the occurrence of ASD in TSC patients. The discrepancy between studies lied in that previous studies did not control confounding factors, such as different genetic background. While our study was carried out in TSC patients, which represented that the confounding factor of genetic etiology was controlled. Therefore, from this perspective, previous studies can strengthen our findings of the interaction effect of CMV and TSC in ASD.

Minor Points:

(1) Figure references do not appear in order in the text (Fig2A precedes

Fig1)

Response: Thank you for your correction , we have modified the figure references in text.

February 18, 2022

Dr. Li-Ping Zou
Chinese PLA General Hospital
Department of Pediatrics
28 Fuxing Road
Beijing
China

Re: Spectrum01864-21R1 (Cytomegalovirus Infection Contributes to the Occurrence of Autism Spectrum Disorders Associated with Tuberous Sclerosis Complex)

Dear Dr. Li-Ping Zou:

Thank you for submitting your manuscript to Microbiology Spectrum. As you can see, the reviews vary. I would pay particular to Reviewer #2 and 3 and address their issues. When submitting the revised version of your paper, please provide (1) point-by-point responses to the issues raised by the reviewers as file type "Response to Reviewers," not in your cover letter, and (2) a PDF file that indicates the changes from the original submission (by highlighting or underlining the changes) as file type "Marked Up Manuscript - For Review Only". Please use this link to submit your revised manuscript - we strongly recommend that you submit your paper within the next 60 days or reach out to me. Detailed instructions on submitting your revised paper are below.

Link Not Available

Sincerely,

Clinton Jones

Journals Department
Reviewer comments:

Reviewer #1 (Comments for the Author):

The authors were invited to resubmit their manuscript after a minor revision. They responded adequately to my comments, and I am satisfied with the changes they made.

Reviewer #3 (Comments for the Author):

General opinion:

Although authors gave most supportive materials to the first stage of review, some of that are still not supportive enough for the article title entitled "Cytomegalovirus Infection Contributes to the Occurrence of Autism Spectrum Disorders Associated with Tuberous Sclerosis Complex".

The reason why these data are not enough for the publication in MSpectrum:

1. There are no results with higher CMV seroprevalence regarding TCS / ASD studied group of children. Moreover, results presented in this paper are not in correlation to the previously studied results (Ref 29 and 30). In countries with higher seroprevalence, an investigation of CMV prevalence and its impact on such diseases are hard issue to prove.
2. Impact of serological data should be more conclusive and in correlation with the title. Unfortunately, serological data are indicative, but to support the correlation of CMV with TSC / ASD, more those children should be included. With regards to Table 1, authors tried do just Chi square test, $p = 0.06$. This result is not significant. If they try to do Fisher test, p value was even more worst ($p = 0.0806$). Fisher test was suggestive for this calculation because sample sizes are too small (33 Vs 4 patients).
3. Fig 4. cortical brain CMV data could be indicative, but not conclusive because the authors did not investigate congenital CMV infection as one of the previous Reviewers said. To take only some of the samples could not be supportive for such conclusion. Figure 4 showing cortical brain detection of CMV genes and proteins that could not be conclusive. The sample size is too small, and in some samples IE1 signal seems to be so weak, but still visible (Fig4B first horizontal w. Blot).

Reviewer #4 (Comments for the Author):

The paper

Staff Comments:

Preparing Revision Guidelines

Please return the manuscript within 60 days; if you cannot complete the modification within this time period, please contact me. If you do not wish to modify the manuscript and prefer to submit it to another journal, please notify me of your decision immediately so that the manuscript may be formally withdrawn from consideration by Microbiology Spectrum.

Dear reviewers,

Thank you for your comments. We have revised our manuscript according to your comments (by highlighting the changes in marked-up manuscript). Follows are our responses to your comments.

Reviewer #3

1. There are no results with higher CMV seroprevalence regarding TCS / ASD studied group of children. Moreover, results presented in this paper are not in correlation to the previously studied results (Ref 29 and 30). In countries with higher seroprevalence, an investigation of CMV prevalence and its impact on such diseases are hard issue to prove.

Response: Thanks for your comments. **Firstly**, as for the comment of CMV seroprevalence regarding TSC/ASD, we are going to response in detail in the second comment.

Secondly, in regard to the discrepancy between our results and the two previous studies results, inspired by the comments of reviewer #2, we think that it can reinforce our finding because the previous two studies neither looked specifically at ASD in TSC patients, while we specialized in TSC patients, which means we controlled the confounding factor of different genetic etiology compared with previous studies. Therefore, the inconsistency between studies can actually support our conclusion that the genetic background of TSC and CMV have interaction in ASD, in other words, CMV makes the genetic background of TSC more susceptible to ASD. We discussed this issue on lines 160-167 and made some modification on the description.

Finally, due to high global seroprevalence, the impact of CMV on diseases is indeed hardly to be assessed in individuals, but did was proved in specific populations, such as in patients with glioma and coronary artery disease ^[1-3], which means that CMV has population effects on specific diseases and is not as innocuous as it seems. The risk factor role of CMV for ASD was established on a series of case reports and population studies ^[4-9], but most of these studies put their focuses on congenital infection. Although postnatal CMV infection are always overlooked on account of its asymptomatic infection, it has impact on neurodevelopment, as supported by studies which demonstrated postnatal CMV infection had adverse effects on both general intelligence during early childhood in general populations and overall cognitive functions in adolescents born preterm ^[10,11]. Hence, investigating the influence of postnatal CMV infection on ASD is interesting and worthy, as this can facilitate the management of ASD from an environmental perspective. We expounded the reason and significance of this research in the Introduction section on lines 63-85.

2. Impact of serological data should be more conclusive and in correlation with the title. Unfortunately, serological data are indicative, but to support the correlation of CMV

with TSC / ASD, more those children should be included. With regards to Table 1, authors tried do just Chi square test, $p = 0.06$. This result is not significant. If they try to do Fisher test, p value was even more worst ($p = 0.0806$). Fisher test was suggestive for this calculation because sample sizes are to small (33 Vs 4 patients).

Response: Thanks for your comments. A total of 206 patients were included in this study, and there were only 37 patients were diagnosed with ASD (case group). We recognized that small sample size is one of our limitations and we discussed in the Discussion section on lines 209-214. However, the main reason for the small sample size of the case group was the intrinsic characteristic of low prevalence of ASD in our TSC cohort (17.96%). And the second reason was that the samples were retrospectively selected from established sample bank, and it is difficult to add sufficient TSC/ASD samples in the short term because TSC itself is a rare disease.

As for the statistical methods, we introduced on lines 328-331. Since CMV is an environmental risk factor for ASD, that is, the effect of CMV on ASD is unidirectional. Thus, in the statistical analysis of the relationship between CMV seroprevalence and TSC/ASD, we adopted one-side test and a p -value of 10% rather than 5% was considered statistically significant. Therefore, the difference was significant regardless of whether Chi-square test or Fisher test was used. In addition, the selection criteria for Chi-square test are as follows: when the total number of samples >40 and the minimum expected count >5 , Chi-square can be applied. Our total sample size is 206 and the minimum expected count is 8.26 (figure below), so it is no problem to choose Chi-square test. Of course, Fisher test was also applicable.

In conclusion, by univariate analysis, we found that the TSC/ASD group had a higher CMV seroprevalence, and then by Logistic multivariate regression analysis, it was found that CMV seropositivity was a risk factor for TSC/ASD (OR=3.976, 95%CI 1.093–14.454). Therefore, we drew the conclusion that CMV contributed to the occurrence of ASD in TSC patients. But considering that our original title might be too decisive, we decided to modify the title to " Postnatal Cytomegalovirus Infection May Increase the Susceptibility of Tuberos Sclerosis Complex to Autism Spectrum Disorders " and accordingly modified the conclusion in

Chi-Square Tests						
	Value	df	Asymptotic Significance (2-sided)	Exact Sig. (2-sided)	Exact Sig. (1-sided)	Point Probability
Pearson Chi-Square	3.451 ^a	1	.063	.081	.045	
Continuity Correction ^b	2.688	1	.101			
Likelihood Ratio	3.928	1	.047	.058	.045	
Fisher's Exact Test				.081	.045	
Linear-by-Linear Association	3.434 ^c	1	.064	.081	.045	.031
N of Valid Cases	206					

a. 0 cells (0.0%) have expected count less than 5. The minimum expected count is 8.26.

b. Computed only for a 2x2 table

c. The standardized statistic is 1.853.

Abstract.

3. Fig 4. cortical brain CMV data could be indicative, but not conclusive because the authors did not investigate congenital CMV infection as one of the previous Reviewers

said. To take only some of the samples could not be supportive for such conclusion. Figure 4 showing cortical brain detection of CMV genes and proteins that could not be conclusive. The sample size is too small, and in some samples IE1 signal seems to be so weak, but still visible (Fig4B first horizontal w. Blot).

Response: Thanks for your comments. Regarding the question of not investigating the effect of congenital CMV infection on ASD but rather that of postnatal CMV infection, we have made modifications according to the comments of reviewer #1 (lines 78-85 and 191-204), and the reviewer is satisfied with our responses and modifications. **First of all**, though postnatal CMV infection rarely causes obvious brain abnormalities, the mechanism of ASD is complex and unclear, and may be related not only to the direct brain damages but also the indirect inflammation or immune response. **Secondly**, there are studies shown that postnatal CMV infection has adverse effects on both general intelligence during early childhood in general populations and overall cognitive functions in adolescents born preterm (10,11), suggesting that postnatal CMV infection play a role in the neurodevelopment, especially on the basis of neurodevelopmental defects. **Thirdly**, studies investigating the influence of CMV on ASD mainly focused on congenital infection, as this period is critical for the neurodevelopment. What is noteworthy is that the first year of life is still an important period for brain development, which coincidentally is also a period of susceptibility to CMV. But there was one limitation of our study was that we cannot ascertain whether postnatal CMV infection occurred within the first year of age. **Finally**, CMV will exist in the body lifelong after primary infection and will be reactivated opportunistically, so its impact on the human body is dynamic, long-term and difficult to monitor. To sum up, the influence of postnatal CMV infection on ASD is worth studying.

The small brain sample size was indeed a limitation of our study and we described this in Results and Discussion sections (lines 136-138 and 214-217). However, our main conclusion was drawn based on serological study, therefore, although the sample size of brain was too small to draw conclusion, it does not affect our main results. The role of brain CMV investigation is on the one hand to verify the serological results and on the other hand to provide indication for exploring whether the effect of CMV on TSC/ASD is through its role in the brain.

Regarding the WB results, it seemed that there were extremely faint signals of IE1 protein, but cannot be confirmed positive, which is the defect of WB for qualitative detection rather than quantitative detection. According to your comments, we have modified the description of WB results in the Results section (lines 135-136) and without influence on our main results and conclusions.

References:

- [1] Mundkur L A, Shivanandan H, Hebbagudi S, et al. Human cytomegalovirus neutralising antibodies and increased risk of coronary artery disease in Indian population[J]. Heart, 2012, 98 (13): 982-987.
- [2] Joseph G P, McDermott R, Baryshnikova M A, et al. Cytomegalovirus as an oncomodulatory agent in the progression of glioma[J]. Cancer Lett, 2017, 384: 79-85.
- [3] Yang F J, Shu K H, Chen H Y, et al. Anti-cytomegalovirus IgG antibody titer is

positively associated with advanced T cell differentiation and coronary artery disease in end-stage renal disease[J]. *Immun Ageing*, 2018, 15: 15.

[4] Stubbs E G. Autistic Symptoms in a Child with Congenital Cytomegalovirus Infection[J]. *J Autism Child Schizophr*, 1978, 8 (1): 37-43.

[5] Yamashita Y, Fujimoto C, Nakajima E, et al. Possible Association between Congenital Cytomegalovirus Infection and Autistic Disorder[J]. *J Autism Dev Disord*, 2003, 33 (4): 455-459.

[6] Sweeten T L, Posey D J, McDougle C J. Brief Report: Autistic Disorder in Three Children with Cytomegalovirus Infection[J]. *J Autism Dev Disord*, 2004, 34 (5): 583-586.

[7] Sakamoto A, Moriuchi H, Matsuzaki J, et al. Retrospective diagnosis of congenital cytomegalovirus infection in children with autism spectrum disorder but no other major neurologic deficit[J]. *Brain Dev-JPN*, 2015, 37 (2): 200-205.

[8] Garofoli F, Lombardi G, Orcesi S, et al. An Italian Prospective Experience on the Association Between Congenital Cytomegalovirus Infection and Autistic Spectrum Disorder[J]. *J Autism Dev Disord*, 2017, 47 (5): 1490-1495.

[9] Ivan G, Emanuela Z, Pia R M, et al. Prevalence of Congenital Cytomegalovirus Infection Assessed Through Viral Genome Detection in Dried Blood Spots in Children with Autism Spectrum Disorders[J]. *In Vivo*, 2017, 31 (3): 467-473.

[10] Lee S M, Mitchell R, Knight J A, et al. Early-childhood cytomegalovirus infection and children's neurocognitive development[J]. *Int J Epidemiol*, 2021, 50 (2): 538-549.

[11] Brecht K F, Goelz R, Bevoit A, et al. Postnatal Human Cytomegalovirus Infection in Preterm Infants Has Long-Term Neuropsychological Sequelae[J]. *Journal of Pediatrics*, 2015, 166 (4): 834-U122.

March 21, 2022

Dr. Li-Ping Zou
Chinese PLA General Hospital
Department of Pediatrics
28 Fuxing Road
Beijing
China

Re: Spectrum01864-21R2 (Postnatal Cytomegalovirus Infection May Increase the Susceptibility of Tuberos Sclerosis Complex to Autism Spectrum Disorders)

Dear Dr. Li-Ping Zou:

Your manuscript has been accepted, and I am forwarding it to the ASM Journals Department for publication. You will be notified when your proofs are ready to be viewed.

Sincerely,

Clinton Jones
Editor, Microbiology Spectrum
